# Impact of COVID-19 Pandemic on Initiation of Immunosuppressive Treatment in Immune-Mediated Inflammatory Diseases in Austria: A Nationwide Retrospective Study

**DOI:** 10.3390/jcm11185308

**Published:** 2022-09-09

**Authors:** Maximilian Kutschera, Valentin Ritschl, Berthold Reichardt, Tanja Stamm, Hans Kiener, Harald Maier, Walter Reinisch, Bernhard Benka, Gottfried Novacek

**Affiliations:** 1Department of Internal Medicine III, Division of Gastroenterology and Hepatology, Medical University of Vienna, 1090 Vienna, Austria; 2Institute of Outcomes Research, Center for Medical Statistics, Informatics and Intelligent Systems, Medical University of Vienna, 1090 Vienna, Austria; 3Ludwig Boltzmann Institute for Arthritis and Rehabilitation, 1090 Vienna, Austria; 4Austrian Health Insurance Fund, 7000 Eisenstadt, Austria; 5Department of Internal Medicine III, Division of Rheumatology, Medical University of Vienna, 1090 Vienna, Austria; 6Department of Dermatology, Medical University of Vienna, 1090 Vienna, Austria; 7AGES—Austrian Agency for Health and Food Safety Ltd., Division of Public Health, 1220 Vienna, Austria

**Keywords:** Immune-mediated inflammatory diseases, conventional immunosuppressive treatment, advanced targeted therapy, inflammatory bowel disease, rheumatoid arthritis, psoriasis

## Abstract

Objective: Conventional immunosuppressive and advanced targeted therapies, including biological medications and small molecules, are a mainstay in the treatment of immune-mediated inflammatory diseases (IMID). However, the COVID-19 pandemic caused concerns over these drugs’ safety regarding the risk and severity of SARS-CoV-2 infection. Thus, we aimed to assess the impact of the COVID-19 pandemic on the initiation of these treatments in 2020. Study Design and Setting: We conducted a population-based retrospective analysis of real-world data of the Austrian health insurance funds on the initiation of conventional immunosuppressive and advanced targeted therapies. The primary objective was to compare the initiation of these medications in the year 2020 with the period 2017 to 2019. Initiation rates of medication were calculated by comparing a certain unit of time with an average of the previous ones. Results: 95,573 patients were included. During the first lockdown in Austria in April 2020, there was a significant decrease in the initiations of conventional immunosuppressives and advanced targeted therapies compared to previous years (*p* < 0.0001). From May 2020 onwards, numbers rapidly re-achieved pre-lockdown levels despite higher SARS-CoV-2 infection rates and subsequent lockdown periods at the end of 2020. Independent from the impact of the COVID-19 pandemic, a continuous increase of starts of advanced targeted therapies and a continuous decrease of conventional immunosuppressants during the observation period were observed. Conclusions: In IMID patients, the COVID-19 pandemic led to a significant decrease of newly started conventional immunosuppressive and advanced targeted therapies only during the first lockdown in Austria.

## 1. Introduction

In December 2019, the Coronavirus Disease 2019 (COVID-19), caused by the severe acute respiratory syndrome coronavirus type 2 (SARS-CoV-2), was first reported in Wuhan, China [1,2,3] from where it rapidly spread throughout the world, leading to a pandemic [4]. As of 30 June 2022, SARS-CoV-2 has affected around 560 million identified cases, with over six million confirmed deaths [5].

While most cases of COVID-19 are mild and have a favorable course, the disease can become severe, resulting in hospitalization, respiratory failure, or even death [6]. The most important reported risk factors for a severe course of COVID-19 are older age, cardiovascular and chronic pulmonary diseases, obesity, diabetes, and immune deficiency [7,8,9].

Conventional immunosuppressive and especially advanced targeted therapies (ADT), including biological medications and small molecules, are a mainstay in the medical treatment of immune-mediated inflammatory diseases (IMID), such as inflammatory bowel diseases (IBD), rheumatic diseases, and psoriasis, as well as less common dermatological inflammatory diseases [10,11,12,13,14]. These medications may be associated with a generally increased risk of infections, such as serious and opportunistic infections described in IBD patients treated with immune-suppressive regimens [15,16] Thus, the COVID-19 pandemic and the speed of its spread caused concerns over the safety of these drugs owing to the lack of evidence respective of risk and severity of infection with SARS-CoV-2. However, early expert consensus balanced out a potentially increased risk of severe COVID-19 by the benefits of continuation of an effective biological treatment, and it was recommended not to stop effective medication [17,18,19,20]. This management of continuation of effective maintenance therapy has also been described in the real-world setting [21].

Little is known about the initiation of conventional immunosuppressive therapies and ADT in IMID during the COVID-19 pandemic. Studies on IBD patients revealed that around 80% to 90% of patients needing to start any biological therapy received their treatment start regularly during the first lockdown [22,23]. On the one hand, pandemic mitigation strategies might have led to the cancelation or postponement of face-to-face meetings with new patients [22,23]. On the other hand, concerns about a potentially increased risk for severe COVID-19 might have contributed to a delay in initiating immunosuppressive and biological drugs. In particular, corticosteroids, methotrexate, azathioprine/6-mercaptopurine, JAK-inhibitors, and rituximab have been mentioned to be associated with an increased risk of severe COVID-19 outcomes [24,25,26]. However, this has not been described at the beginning of the pandemic. A survey among the European Alliance of Associations for Rheumatology countries revealed a delay between symptom onset and a first rheumatological visit as well as the postponement of treatment decisions during the first wave of the COVID-19 pandemic, which negatively impacted early treatment and treat-to-target strategies requiring tight control [27]. Such undertreatment could lead to flares and complications of the underlying inflammatory disease with subsequent hospitalizations. From a macro-level perspective, especially the initiation of new ADT could be an indicator for estimating undertreatment in patients with IMID.

Therefore, our nationwide study aimed to assess the number of newly started conventional immunosuppressive and advanced targeted therapies during the first waves of the COVID-19 pandemic in the year 2020 in Austria and to compare these data with the respective timeframe of previous years.

## 2. Materials and Methods

### 2.1. Design

Based on dispensing data from Austrian health insurance funds, we conducted a population-based retrospective analysis with a four-year observation period from 2017 to 2020. Dispensing data means that all data on prescribed medications picked up at the pharmacy and covered by the Austrian health insurance funds can be retrieved. Austrian health insurance funds cover 98% of all residents in Austria (8,755,124 persons in December 2020).

### 2.2. Participants and Data Extraction

Data from all patients with initiations of conventional immunosuppressive therapy, ADT, including biological medications and small molecules, and other disease-specific medications were included. The initiation of treatment was defined as all first prescriptions from 2017 to 2020. To fulfill this definition, no previous prescription of the same drug among the listed medications was allowed from the beginning of the previous year (in 2016) (for timeline of the study see Appendix A). However, due to Austrian regulations, only patients who received their medications outside of a hospital could be included since treatment data of hospitalized patients were not available for our analysis. The diagnosis of the IMID could only be recorded in case of hospitalization during the observation period, so we examined new prescriptions of drugs approved and reimbursed for IMID in Austria, despite being unaware of the patients’ exact diagnosis. To increase the number of known diagnoses, we assigned medications to a diagnosis of an IMID if the medication was approved only for that indication.

### 2.3. Objectives

The primary objective was to evaluate the impact of the COVID-19 pandemic on the initiation of conventional immunosuppressive therapy and ADT in the year 2020 compared to the period 2017 to 2019 in IMID in Austria. The secondary objective was to evaluate the course of initiation of these treatments in IMID in Austria during the observation period regardless of the COVID-19 pandemic.

The primary endpoint was the first prescription of conventional immunosuppressive therapy, ADT, and other medications for immune-mediated inflammatory diseases.

### 2.4. Data Analysis

The frequency of newly started conventional immunosuppressive therapy, ADT, and other specific medications was calculated for every month of 2020 and compared with monthly prescription rates of the three previous years (2017–2019). The medications used for this analysis were categorized and are listed in Table 1. Other specific medications without immunosuppressive effects were included as a possible sign of hindered contact between patients and physicians due to pandemic mitigation strategies.

Re-identifying subjects by the Medical University of Vienna was impossible as only birth year, high-level region of residence (one of the nine Austrian counties), gender, and, if applicable, death year were included. Moreover, ethical approval was given by the ethics committee and internal review board of Vienna (EC No. 1330/2021).

### 2.5. Statistical Analysis

Data sets from 13 different health insurances were combined. Within this step, we adjusted for multiply insured individuals by combining their data. For categorical data, absolute and relative frequencies were calculated and depicted using bar charts. Metric variables were summarized by calculating the mean and standard deviation. This was done for the entire study sample and separately for subgroups.

Differences between groups were tested using Chi-2 tests (categorical data), *t*-tests, or Wilcoxon–Mann–Whitney tests (metric data). To balance the potential effects of oversampling, we assessed the clinical meaningfulness of significant differences. Medication categories were summarized where appropriate. Initiation rates of medication were calculated by comparing a certain unit of time (month, year) with an average of the previous ones. *p*-values below 0.05 were considered statistically significant. We used R (https://www.r-project.org, R version 4.2.1, accessed on 23 June 2022)) to perform the statistical analysis.

## 3. Results

### 3.1. Patients’ Characteristics

We identified 95,573 patients with the start of at least one conventional immunosuppressive and/or advanced targeted therapy and/or other disease-specific medication from 2017 to 2020. Of these, 43,402 were male (45%; mean age 50.6 years; SD ± 18.1 years), and 52,171 were female (55%; mean age 53.4 years; SD ± 18.7 years). The diagnosis of IMID recorded in case of hospitalization was available in 9.7% of the patients (Table 2). In addition, in a total of 59.5% patients, we could assign diagnoses by using medications that were only approved for a single indication during the observation period (Appendix A). In the entire data set, 122,213 medications were started in the years 2017 to 2020. The number of starts of every medication is given in the Appendix A. The majority of the patients had only one treatment start (90,021; 94.2%), 3822 patients (4.0%) had two, 1500 patients (1.6%) had three, and 230 patients (0.2%) had four or more treatment starts with different medications.

### 3.2. Impact of the COVID-19 Pandemic on the Initiation of Conventional Immunosuppressive and Advanced Targeted Therapies

During the first lockdown in Austria in spring 2020 (week 12–week 20), there was a significant decrease in the overall initiation of conventional immunosuppressive therapies and ADT in April 2020 compared to previous years (all *p* < 0.0001, one sample Chi-2 tests) (Figure 1, Appendix A). After the first lockdown, initial prescriptions of conventional immunosuppressive therapies and ADT re-achieved pre-lockdown levels despite higher infection rates with SARS-CoV-2 in the total population. In addition, in subsequent lockdown periods (second lockdown in Austria week 45–week 48; third lockdown week 53; Figure 2, the frequency of initiation of conventional immunosuppressive therapies and ADT did not decrease again (Figure 1 and Figure 3). This was mainly observed for the starts for biologics which even exceeded the number of starts in the corresponding months of the previous years (Figure 1, Appendix A).

We also investigated the initiation of mesalazine (Appendix A), which was also significantly reduced in April 2020 during the first lockdown compared to the corresponding months of the previous years (both *p* < 0.0001). No mesalazine initiation changes occurred during subsequent lockdowns at the end of 2020.

### 3.3. Changes in the Initiation of Therapies during the Observation Period Independent from COVID-19

Independent from the COVID-19 pandemic, the number of initiations of different medication groups significantly changed during the observation period (2017–2020). We detected a continuous rise in the initiation of biological medications of 6.7% per year as well as of small molecules of 6.1% per year, respectively (calculated as compound annual growth rate, both *p* < 0.0001) (Figure 2). The starts of small molecules increased from 2017 to 2018 by 22.8% and stayed stable in the following years. In contrast, the start of immunosuppressive medications significantly decreased by 5.9% per year (*p* < 0.0001) (Figure 2), and mesalazine even by 6.1% per year (*p* < 0.0001) (Appendix A).

## 4. Discussion

Our nationwide study revealed that the COVID-19 pandemic led to a significant decrease in the initiation of conventional immunosuppressive and advanced targeted therapies in patients with immune-mediated inflammatory diseases, including inflammatory bowel diseases, during the first lockdown in 2020 in Austria. However, after that, the initiation of these substances rapidly re-achieved pre-lockdown levels despite much higher infection rates with SARS-CoV-2 and subsequent lockdown periods at the end of 2020.

Little has been reported about the initiation of conventional immunosuppressive medications and ADT during lockdown periods. An Italian web-based survey revealed that 79% to 92% of patients needing to start any intravenous or subcutaneous biological therapy, respectively, received their first administration regularly during the first lockdown [22]. Another study reported that biological therapy started as planned in 21 patients out of 25 (86%) [23]. However, as far as we are aware, no data have been published about either the initiation of other IMID-specific medications during the first lockdown period or about the initiation during subsequent lockdowns resulting from even higher infection rates.

That decrease in the initiation of therapy during the first lockdown seemed to be caused by the concern that conventional immunosuppressive medications and ADT could increase the risk of infection with SARS-CoV-2 and a more severe course of COVID 19 in case of an infection. However, for patients with immune-mediated inflammatory diseases, an increased risk of infection with SARS-CoV-2 and a more aggressive course has not been demonstrated in several later publications [28,29,30,31,32], though the literature also revealed partially conflicting results [33,34,35,36]. Some medications have been associated with an enhanced risk of severe COVID-19 outcomes. This has been described for corticosteroids, methotrexate, azathioprine/6-mercaptopurine, JAK-inhibitors, and rituximab, but not for the other in IMID broadly used biologics [20,37,38,39,40]. This gain in knowledge and subsequent recommendations of national and international scientific societies is likely to be mainly responsible for the lack of impact of high infection rates with SARS-CoV-2 and subsequent lockdowns on start of immunosuppressive and biological medications at the end of 2020 [41].

Another probable reason for the reduced initiations of immunosuppressive and biological medications might have been reduced contact between patients and physicians due to pandemic mitigation strategies, which might have led to the cancellation or postponement of face-to-face meetings [22,23]. This assumption might be confirmed by the fact that the number of starts with mesalazine, which does not seem to have any immunosuppressive effect, was also significantly reduced during the first lockdown, which might be a sign of reduced patient visits during the lockdown. Consistent with this observation, effects in other fields of patient care have been described. For example, it was reported that back in the spring of 2020, the pandemic led to a marked reduction in the number of people referred, diagnosed, and treated for colorectal cancer [20,42]. In addition, other countries also observed a significant decrease in IBD-related procedures during the first lockdown, especially in April 2020 [43].

Independent of any impact of the COVID-19 pandemic, our study revealed a continuous increase of starts of biological medications and small molecules during the observation period from 2017 to 2020. Other publications have reported similar findings over the last years [43,44,45,46,47]. Biosimilars led to overall reductions in health care expenses. On the other hand, the start of conventional immunosuppressive drugs decreased continuously during the same period. The increase in biological medications and small molecules could also be due to an increasing incidence and prevalence of immune-mediated inflammatory diseases. Still, corresponding data are not available for Austria. However, the decreasing number of starts of immunosuppressive medications that are already longer on the market suggests that it is more likely to shift to a more progressive treatment attitude according to recommendations of scientific organizations [48,49,50,51].

We did not observe a decrease in immunosuppressive and biological treatments in the entire year 2020. This means that the start of the immunosuppressive and biological medications was only delayed for a short period.

Interestingly, there was also a significant annual decrease in the start of mesalazine during the observation period. Mesalazine is only approved for inflammatory bowel diseases. It is a mainstay in treating mild to moderate ulcerative colitis but has low effectiveness in Crohn’s disease. As we do not have any reason to believe that the prevalence of ulcerative colitis decreased within the last years in Austria, this finding is likely due to diminishing prescriptions for Crohn’s disease [52]. As there has been a report that the usage of mesalazine could be associated with an increased risk of severe COVID-19 [25], this was still unknown at the time of the first wave of the pandemic and could not be confirmed later on [37]. Therefore, we assume that this had no influence on the prescription of mesalazine during the first wave of COVID-19.

The study has its strengths but also limitations. The data of the Austrian health insurance funds cover the majority (98%) of the Austrian population. Furthermore, we included all starts for immunosuppressive and biological treatments and small molecules, mainly in IBD, rheumatologic diseases, and psoriasis. However, only medications prescribed on an outpatient basis could be recorded since treatment data of hospitalized patients were not available for our analysis. This might have affected medications that are given intravenously, e.g., infliximab. Nevertheless, as some of these drugs were used off-label in patients with COVID-19, the presence of inpatient treatments might have been a bias and therefore, in the absence of inpatient treatment, there can be no prescriptions due to COVID-19 disease. However, we assume that the overall results are barely affected as the vast majority of patients are being prescribed ADT during ambulatory visits. In this analysis, we focused on prescribed first courses of ADT only. To determine the pattern of use of other medications, including for example corticosteroids or other medications such as budesonide and beclomethasone, would be an interesting further analysis. It would require a different dataset extracted also from the electronic health records with information on discontinuation, changes in dosing, patients not taking medications, etc. A further limitation is that we could not assign our findings to specific diagnoses since the diagnosis recorded in case of hospitalization was only available in around 10% of the patients. However, if we assigned medications to only approved indications, the percentage of available diagnoses would rise to approximately 60%. We considered comparing the number of initial courses of a drug to other substances to assess whether drugs with greater a priori safety, such as ustekinumab or vedolizumab, were prescribed more often compared to drugs which were considered to have a slightly higher risk of infection, such as TNF-alpha inhibitors. However, as the number of cases for each drug was too small, we could not provide for a reliable statement.

## 5. Conclusions

In summary, our study revealed a significant decrease in initiations of conventional immunosuppressive and advanced targeted therapies, including biological treatments and small molecules, during the first COVID-19 pandemic lockdown in April 2020. Concerns over an increased risk of infection with SARS-CoV-2 and a more severe course of COVID-19 in case of an infection and pandemic mitigation strategies with subsequent cancellation or postponement of face-to-face meetings appear to be the most important reasons for that finding. However, that decrease in initiations of medications was not observed during subsequent lockdown periods despite much higher infection rates. One can guess that this was also true in later periods with high infection rates in 2021 and at the beginning of 2022, encouraged by the availability of vaccinations against SARS-CoV-2 as the most effective option to prevent severe COVID-19. Independent of the impact of the COVID-19 pandemic, we observed a continuous increase in the start of biological medications and small molecules and a continuous decrease of conventional immunosuppressants during the observation period from 2017 to 2020.

## Figures and Tables

**Figure 1 jcm-11-05308-f001:**
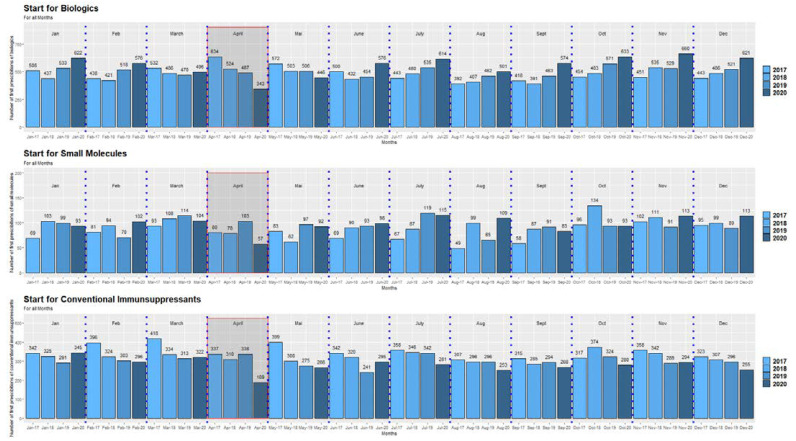
Starts of biological medications, small molecules, and conventional immunosuppressive medications in the period 2017–2020. The red rectangles mark the timeframe of a significant decrease of the initiation of biologics, small molecules as well as conventional immunosuppressants during the first lockdown (April 2020 compared to the corresponding timeframe of previous years). The scaling of the *y*-axis differs for the presentation of the three medication groups to demonstrate at best the decrease of the medication starts in April 2020.

**Figure 2 jcm-11-05308-f002:**
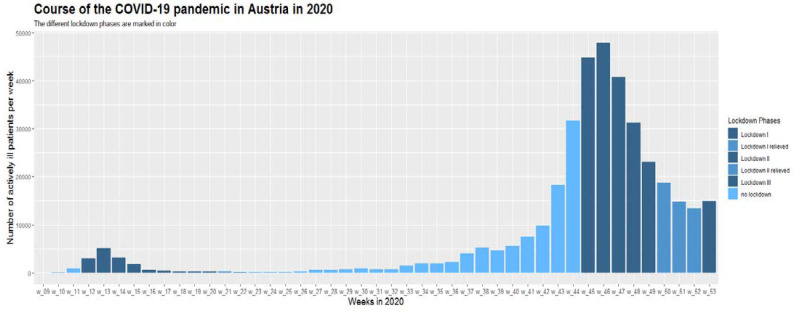
Positive COVID-19 cases by week in 2020 in Austria (first positive laboratory diagnosis).

**Figure 3 jcm-11-05308-f003:**
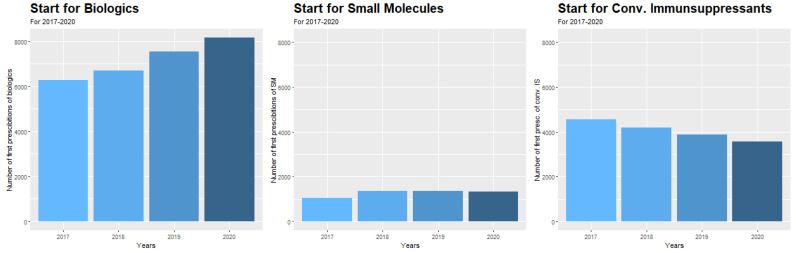
Annual starts for biological medications, small molecules, and conventional immunosuppressive medications in the period 2017–2020.

**Table 1 jcm-11-05308-t001:** Advanced targeted therapies (biologics and small molecules) and conventional immunosuppressive medications, and other specific medications included in the analysis.

Biologics	TNF-Alpha Inhibitors	Adalimumab, Certolizumab Pegol, Etanercept, Golimumab, Infliximab
	Anti C5	eculizumab
	IL-1 inhibitors	anakinra, canakinumab
	IL-4 inhibitors	dupilumab
	IL-6 inhibitors	sarilumab, tocilizumab
	IL-17 Inhibitors	brodalumab, ixekizumab, secukinumab
	IL-23 inhibitors	guselkumab, risankizumab, tildrakizumab
	IL12/23 inhibitor	ustekinumab
	Anti-BAFF	belimumab
	B-cell depletion	rituximab
	integrin α_4_β_7_ inhibitor	vedolizumab
	Co-Stimulation inhibitor	abatacept
Small molecules	PDE4 inhibitors	apremilast
	JAK-inhibitors	baricitinib, tofacitinib, upadacitinib
Conventional immunosuppressive medications		azathioprine, cyclosporine, leflunomide, mycophenolate mofetil, methotrexate
Others		sulfalazine, mesalazine

**Table 2 jcm-11-05308-t002:** Diagnoses of immune-mediated inflammatory diseases in 9234 patients.

Diagnosis	n (%)
Crohn’s disease; n (%)	3488 (37.8)
Ulcerative colitis; n (%)	2805 (30.4)
Rheumatoid arthritis; n (%)	1543 (16.7)
Plaque psoriasis; n (%)	629 (6.8)
Ankylosing spondylitis; n (%)	259 (2.8)
Hidradenitis suppurativa; n (%)	179 (1.9)
Childhood arthritis; n (%)	151 (1.6)
Uveitis; n (%)	112 (1.2)
Psoriatic arthritis; n (%)	35 (0.4)
Behçet–Krankheit; n (%)	33 (0.4)

## Data Availability

The dataset and R code can be obtained from the corresponding author upon reasoned request.

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
