# Peer review of "Impact of COVID-19 Pandemic on Initiation of Immunosuppressive Treatment in Immune-Mediated Inflammatory Diseases in Austria: A Nationwide Retrospective Study"

_jcm, 2022, doi:10.3390/jcm11185308_

Round 1
Reviewer 1 Report
This is a nationwide retrospective study which showed the reduced use of newly started conventional immunosuppressants and advanced targeted therapies (ADT) only during the first lock down due to COVID-19 in Austria. This is an interesting and reasonable report as many data have supported that most of ADT do not necessarily increase the risk of COVID-19 infection. On the other hand, a meta-analysis published in the early phase of COVID-19 pandemic suggest that steroid use is supposed to be associated with an increased risk of COVID-19 infection (PMID 33051220). This reviewer has some comments.
1. In the Introduction, the authors mentioned medications for IMIDs may be associated with an increased risk of infections [15, 16] (Lines 54-55). However, studies of citations 15, 16 did not assess the risk of COVID-19 in patients with IMIDs. This reviewer would like to suggest that the authors can introduce “steroids, MTX, AZA/6-MP, JAK, rituximab may increase the risk of COVID-19”, which was described in the Discussion section (Lines 205-208). In the Introduction, the recent data on the association of COVID-19 infection risk and medications for IMIDs need to be fully introduced for the readers.
2. In terms of Figure S3, the authors found no mesalamine initiation changes occurred during subsequent lockdowns at the end of 2020. They discussed it may be associated with decreased frequencies of in-person visits or diminishing prescriptions for Crohn’s disease. Studies assessing an international registry of IBD published in the early phase of COVID-19 pandemic (PMID 33082265, 32425234) showed mesalamine/sulfasalazine can be associated with an increased risk of severe COVID-19. Do the authors think these data could change the practices of IBD specialists during COVID-19 pandemic? Please discuss.
3. Several studies found systemic steroid use is associated with the risk of COVID-19. So, if the authors can demonstrate the trend of steroid use before and after COVID-19 pandemic (2017-2020), it may strengthen the conclusions of this manuscript. This reviewer presumes that systemic steroid use may have been reducing after the lock-down, in contrast to ADT. Please consider adding data regarding the trend of steroid use for IMIDs, if possible.
Reviewer 2 Report
I read with great interest your paper entitled "Impact of the pandemic". I consider the work to have a very interesting basis, especially in its ability to cover almost the entire population of a country. It is true that, as the authors themselves point out, the inability to assign a diagnosis with certainty is limiting. The lack of hospital treatment is also an important limitation.
However, I believe that the fact of not including hospital treatments may be a strength: some of these drugs were used off-label in patients with COVID 19 (tocilizumab, baricitinib, anakinra). If inpatient treatments had been included there would have been a bias in this respect. I think this fact can be discussed by the authors to make it clear that, in the absence of inpatient treatment, there can be no prescriptions due to COVID-19 disease.
Have the treatments been analysed by groups? It is possible that there has been an increase in drugs with greater a priori safety (for example in IBD, ustekinumab or vedolizumab against anti-TNF) to the detriment of drugs with a higher theoretical risk of infections.
I consider it a very important strength to have mesalazine prescription as a reference. Corticosteroids could have been another control, because of their use in inflammatory pathology, but again their indication for the treatment of COVID precludes their use in this study. However, perhaps the prescription data for oral budesonide and beclomethasone could be analysed to corroborate the data on mesalazine use.
Finally, I would like to know (and if possible for the authors to specify in the text) whether these are prescribing, dispensing or administration data, as it changes the interpretation of the study slightly. Given the drug profile, I imagine that it is dispensing data, but I would appreciate clarification.
In conclusion, I think it is an interesting study, with an appropriate methodology and which allows us to see the overall impact of the pandemic on the use of "immunosuppressive" drugs. I would therefore like to congratulate the authors for their work.
Translated with www.DeepL.com/Translator (free version)
Round 2
Reviewer 1 Report
Thank you for your replies to my comments. I think the authors have appropriately revised their manuscript.